



# Uncertainty model for dual-Doppler retrievals of wind speed and wind direction

Nikola Vasiljević[1], Michael Courtney[1], and Anders Tegtmeier Pedersen[1]

[1]Technical University of Denmark - DTU Wind Energy, Frederiksborgvej 399, Building 118-VEA, 4000 Roskilde, Denmark

**Correspondence:** Nikola Vasiljević (niva@dtu.dk)

**Abstract.** In this paper we present an analytical model for estimating the uncertainty of the horizontal wind speed based on dual-Doppler lidar measurements. The model follows the propagation of uncertainties method and takes into account the uncertainty of radial velocity estimation, azimuth and elevation pointing angles, and ranging. The model is achieved by coupling ranging and elevation angle to uncertainty of the probed wind speed through a simple power-law shear model. The model has been implemented in Python and made freely available through as the Python package *YADDUM*.

## 1 Introduction

### 1.1 From meteorological masts to multi-lidars

In the wind energy domain, due to the economical consequences when developing and operating wind farms, all wind measurements must have a well-defined uncertainty. As an example, for a large offshore wind farm project, considering conservative calculations Hasager et al. (2013), decreasing the uncertainty on the predicted wind resources at wind turbine hub height by 0.1 m/s leads to an estimated saving worth around 10 M£ per year for the 25 year lifetime of the farm. Estimates of wind speed uncertainty are an essential element of planning and obtaining finance for wind energy projects. There are also clear motivations and needs to focus our efforts on understanding what drives the uncertainty in wind measurements and how it can be reduced.

Traditionally, wind measurements have been acquired by met mast based sensors such as cup anemometers Kristensen (1999). These instruments, which have been around for well over 100 years, were a backbone for the rapid development of the wind industry. The uncertainty of a cup anemometer has been thoroughly studied and reduced in the past several decades, driven by the dependency of the wind industry on them.

However, the rapid development of the wind industry brought more powerful and thus taller wind turbines, imposing a need for wind speed measurements at greater and greater heights. Due to the high costs of tall met masts, the traditional wind speed measurements from the heights where modern wind turbines operate became economically unfeasible. In a search for cost-





attractive yet accurate wind speed measurements, over the past two decades the wind energy community started to embrace wind lidars (also known as coherent Doppler lidars or heterodyne Doppler lidars) as an alternative to the mast based sensors.

Wind lidars acquire wind speed information remotely by probing the atmosphere with laser light. Therefore, contrary to the cup anemometers, there is no physical contact with the moving air. The inherent nature of lidars is that they can directly
resolve radial velocity or line-of-sight (LOS) wind speed, thus only a projection of the wind vector on the laser beam direction. In certain flow conditions, this constraint can be overcome by probing the volume of air in several beam directions (see for example Browning and Wexler (1968); Strauch et al. (1987); Cariou and Boquet (2011)) and assuming the horizontal homogeneity of the flow. Wind lidars come either with fixed or with flexible scanning geometry. The first group of lidars includes so-called vertical profilers (mimic met mast measurements, e.g. Courtney et al. (2008)) and horizontal profilers (i.e.,
nacelle-based lidars, e.g. Borraccino et al. (2016)) while scanning lidars (e.g., Vasiljevic et al. (2016)) correspond to the second group.

Detailed studies of single lidars with fixed scanning geometry showed an impressive agreement of the wind speed and direction (i.e., single-Doppler retrievals) with the wind speed and direction measurements from mast based sensors in flat terrain and offshore Courtney et al. (2008); Peña et al. (2008); Borraccino et al. (2016). This does not come as a surprise
since the horizontal homogeneity of the flow, an underlying assumption in the retrieval algorithm, is usually satisfied in such conditions. For fixed scanning geometry lidars operating in homogeneous flows, there is a well-established body of literature covering the metrological aspects of wind speed measurements (e.g., Borraccino et al., 2016).

However, once the flow becomes influenced by the terrain morphology or near-by objects, the assumptions of the horizontal homogeneity of the flow fails, leading to the increased uncertainty of single-Doppler retrievals. This has been investigated
in several studies where vertical profilers were compared with measurements from near-by mast based sensors in complex terrain Bingöl et al. (2009); Bradley et al. (2015). The error encountered in complex terrain is usually unacceptably high for applications in the wind energy domain. In these situations, the best approach is to use multi-lidar instruments, based on two or three synchronized scanning lidars Vasiljevic et al. (2016). Even though triple Doppler solutions are capable of acquiring a full 3D wind vector (e.g., Simley et al. (2016); Vasiljević et al. (2017))), dual Doppler setups are more commonly used (Hill et al.
(2010); Iungo et al. (2013); Newsom et al. (2015)) for several reasons such as ease of operation and cost reduction. Besides tackling high-quality measurements in complex terrain (e.g., Pauscher et al. (2016); Vasiljević et al. (2017)), the multi-lidar approach is an attractive way of performing measurements of near-shore offshore wind resource from the shore-line Floors et al. (2016), since it avoids costly offshore lidar installations.

## 1.2 Multi-lidar metrology

The metrological aspects of multi-lidar measurements have been the topic of several communications. The inter-comparisons between multi-lidar setups and in-situ measurements from mast based sensors show excellent agreement in both flat terrain (dual-Doppler Vasiljevic et al. (2016), triple-Doppler Mann et al. (2009); Berg et al. (2015); Fuertes et al. (2014)) and complex terrain (dual- and triple- Doppler Pauscher et al. (2016); Vasiljevic et al. (2016). As it was shown earlier in the case of multi-radars Davies-Jones (1979), the geometry of the multi Doppler setups (i.e., its layout) can have a tremendous impact on the



uncertainty of the acquired wind speed, since the between-beam angle (i.e., intersecting angle between outgoing beams) can amplify the total uncertainty. In most of the inter-comparison studies, the geometry of the lidar setups with the respect to the in-situ measurements was close to ideal (e.g., Mann et al. (2009); Fuertes et al. (2014) the between-beam angle was 90°). This significantly reduced the potential amplification of the total uncertainty. Taking into account that the laser beam pointing and

ranging was well configured, the resulting uncertainty in the aforementioned studies was mainly driven by the uncertainty in the retrieved LOS speed. For want of a better alternative, the LOS speed uncertainty is obtained through comparisons with a reference cup anemometer and is therefore limited to the cup's uncertainty. In case of multi-lidars, the implication of the geometry was discussed in several publications (e.g., Hill et al. (2010); Stawiarski et al. (2013); Simley et al. (2016); Pauscher et al. (2016); Debnath et al. (2017); Vasiljević et al. (2017)).

Stawiarski et al. (2013) confirmed the same results regarding the amplification properties of the between-beam angle for dual-Doppler setups as those reported in Davies-Jones (1979). The study Pauscher et al. (2016), which included simultaneous measurements with several multi-lidar setups (i.e., different between-beam angles), showed differences in the comparison of these setups with in-situ measurements. The same study suggested that the differences are probably caused by the multi-lidar setup. On the other hand, communications such as Simley et al. (2016) and Debnath et al. (2017) proposed a norm approach

to assess the suitability of the multi-lidar setup before an actual campaign. The actual uncertainty analysis of multi-lidar measurements was to a certain extent the topic of Hill et al. (2010), while a deeper overview of different error sources in multi-lidar measurements was discussed in Stawiarski et al. (2013), whereas Vasiljevic (2014) provided in-depth analysis of contributions to the pointing uncertainty and attempted to develop a total pointing uncertainty model.

The authors in (Hill et al., 2010) studied the uncertainty of dual-Doppler retrievals of the vertical component and the compo-

nent of the horizontal wind speed aligned with the plane of two overlapping range-height indicator (RHI) scans. The uncertainty analysis was limited to the uncertainty contribution originating from the random error in the acquired radial velocity. (Hill et al., 2010) developed a simple uncertainty model by propagating a random error in the acquired radial velocity through the algorithm (see Newsom et al. (2008)) used to convert the independent radial velocity measurements to the two aforementioned components of the wind vector. Afterward, the model was tested numerically by polluting real measurements with random

noise. This study qualitatively reported the robustness of the algorithm described in Newsom et al. (2008) with respect to the between-beam angle. The study communicated in Stawiarski et al. (2013) provided an extensive catalog of different errors associated with lidar measurements, showed their individual impact on the measurement uncertainty and, as we stated earlier, confirmed the previous amplification effects of the between-beam angles.

Despite providing in-depth details of different aspects of the multi-lidar (dual-Doppler) measurement uncertainty, the au-

thors in Stawiarski et al. (2013) did not realize a total uncertainty model, which would entail combining various uncertainty contributions altogether, but treated uncertainty contributions individually. Also, the impact that the pointing uncertainty has on the total measurement uncertainty was analyzed only from the mathematical perspective (i.e., slight differences in the actual projection of the wind vector on the LOS). Consequently, the above study concluded that this uncertainty contributor has a negligible impact on the total uncertainty. This is in fact, usually not correct. The most important impact on the overall un-

certainty that the pointing uncertainty introduces arises not from the incorrect wind vector projection (mathematics) but from





the uncertainty in height at which the radial velocity is acquired. Due to the vertical shear, the uncertainty in height at which measurements take place (as we will see later) translates into a significant contribution to the total uncertainty. Furthermore, the aforementioned study did not include the uncertainty contribution arising from the ranging uncertainty where again, in combination with wind shear, significant wind speed errors can arise since the 'wrong' wind speed will be sensed.

In our review of the literature, we find that there is a general inconsistency in the use of metrological terms (e.g., accuracy misinterpreted as trueness, precision as accuracy, etc.). Also, we noticed a mixture of approaches to assessing uncertainty. Therefore, in our opinion, we see a need for a well-rounded community-driven uncertainty model that includes all the important uncertainty contributors, which is consistent with and communicated following the well-established metrological standards. This was to a large extent highlighted in the latest communication by IEA Wind Task 32 Clifton et al. (2018). With this paper,

we intend to lay out the basis for the development of a community-based uncertainty model.

The paper is organized as follows. We start by the analysis of different uncertainty contributions to the lidar measurement uncertainty. In Section 3 we will model radial velocity uncertainty and the uncertainty of dual-Doppler retrievals of horizontal wind speed and wind direction. The derived models are demonstrated in Section 4. Section 5 discusses results and our future work, whereas Section 6 holds our concluding remarks.

## 15   2   Uncertainty contributions to lidar measurements

As stated in GUM Joint Committee for Guides in Metrology (2008) the objective of a measurement is to determine the value of the measurand, that is, the value of the particular quantity to be measured. Accordingly, any measurement should begin with an appropriate specification of the measurand, the method of measurement and the measurement procedure.

In our case, the measurand is a horizontal wind speed $V_H$ and wind direction $\Theta$, or horizontal components $u_{wind}$ and $v_{wind}$

of the wind vector $V$, in one or several points of the interest in the atmosphere. To measure directly the $u_{wind}$ and $v_{wind}$ components of the wind vector we need a minimum of two wind lidars which can simultaneously retrieve independent radial wind speeds (i.e., $V_{radial1}$ and $V_{radial2}$) at the points of interest. Each lidar acquires the LOS measurement by emitting and steering the laser light towards the points of the interest and detecting the light which is reflected by the moving aerosols particles (i.e., backscattered light) from the locations of interest in the atmosphere. With this information we can establish a

simplified mathematical model of dual-Doppler measurements:

$$u_{wind} = f_{u_{wind}}(V_{radial1}, V_{radial2}) \tag{1}$$

$$v_{wind} = f_{v_{wind}}(V_{radial1}, V_{radial2}) \tag{2}$$

$$V_H = f_{V_H}(V_{radial1}, V_{radial2}) \tag{3}$$





$$\Theta = f_\Theta(V_{rad1}, V_{radial2}) \tag{4}$$

Due to the above-mentioned amplification of uncertainties caused by the Dual-Doppler setup, our procedures rely on optimizing the dual-Doppler setup such that the amplification of the combined uncertainty is acceptable for a given measurement campaign. However, before we proceed to optimize the measurement setup we need to understand what are the most fundamental uncertainties associated with lidar measurements. By analyzing the above-mentioned measurement method essentially two questions arise:

– How accurately one can estimate the radial wind speed from the backscattered light?

– How accurately one can determine from which part of the atmosphere the backscattered light is reflected?

These two questions identify three fundamental sources of uncertainties, which are:

1. Uncertainty in estimating radial wind speed from the backscattered light ($u_{est}$)

2. Pointing uncertainty expressed in terms of uncertainties on azimuth $\theta$ and elevation $\varphi$ angles that constitute the laser beam direction $D$ ($u_\theta$ and $u_\varphi$)

3. Ranging uncertainty ($u_R$)

Based on the previous, we can establish a single lidar measurement model:

$$v_{radial} = f(v_{estimated}, \theta, \varphi, R) \tag{5}$$

where $v_{estimated}$ is LOS speed retrieved by processing the backscattered light and $R$ is the range along a given laser beam direction expressed through the azimuth and elevation angles.

## 3 Deriving the uncertainty models

In the following we derive the uncertainty models for wind speed and wind direction.

### 3.1 Assumptions

To achieve a simple dual-Doppler uncertainty model we are taking into consideration several assumptions. Specifically we assume that:

– Lidars are point measurement devices, thus we don't consider the impact of the probe length on the measurement uncertainty.





- The uncertainty contributors are uncorrelated.

- Lidars will operate with a shallow elevation angle (e.g., below 5°) and that the vertical wind speed is low and can therefore be omitted in the model derivation.

- Wind will only changes with height (horizontal homogeneity) and that it follows a power-law profile.

5    - Positions of lidar and the desired measurement locations are exactly known.

## 3.2 Radial wind speed uncertainty model

Figure 1 depicts a scanning lidar which measures a radial wind speed $V_{radial}$ in a point $M$ by beaming the laser beam (which direction is determined by the elevation $\varphi$ and azimuth angle $\theta$) and resolving the backscattered light at a range $R$ along that beam. Radial wind speed is equal to the projection of the wind vector $\boldsymbol{V}_{\text{wind}}$ on the laser beam. We define the wind vector

10   $\boldsymbol{V}_{\text{wind}}$ as the following:

$$\boldsymbol{V}_{\text{wind}} = (u, v, w) \tag{6}$$

where $u$ is the zonal velocity (i.e., component of the horizontal wind towards the East); $v$ is the meridional velocity (i.e., component of the horizontal wind towards the North); and $w$ is the upward air velocity or simply vertical wind speed.

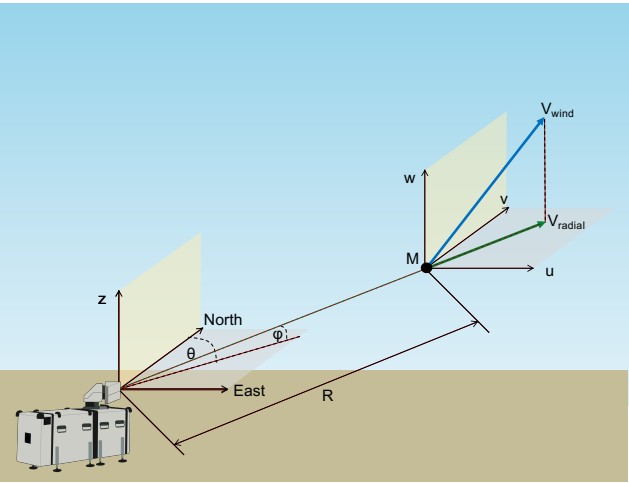

**Figure 1.** Single scanning lidar probing atmosphere: $M$ - probed measurement point, $\theta$ - azimuth angle, $\varphi$ - elevation angle, $R$ - range, $u$ - zonal velocity, $v$ - meridional velocity, $w$ - upward air velocity, $\boldsymbol{V}_{\text{wind}}$ - wind vector and $V_{radial}$ - radial wind speed.





The relation between the wind vector and radial wind speed can be mathematically described in the following way:

$$V_{radial} = \boldsymbol{n}.\boldsymbol{V}_{wind} = \begin{pmatrix} \sin(\theta)\cos(\varphi) \\ \cos(\theta)\cos(\varphi) \\ \sin(\varphi) \end{pmatrix} \cdot \begin{pmatrix} u \\ v \\ w \end{pmatrix} \tag{7}$$

where $\boldsymbol{n}$ is a unit vector which described the laser beam direction. Since we assumed that $\varphi$ is close to zero and that $w$ is low, we can reduce Equation 7 to:

$$V_{radial} = \begin{pmatrix} \sin(\theta)\cos(\varphi) \\ \cos(\theta)\cos(\varphi) \end{pmatrix} \cdot \begin{pmatrix} u \\ v \end{pmatrix} \tag{8}$$

or:

$$V_{radial} = u\sin(\theta)\cos(\varphi) + v\cos(\theta)\cos(\varphi) \tag{9}$$

The horizontal components $u$ and $v$ can be expressed in terms of the horizontal wind speed $V_h$ and wind direction $\Theta$ as following:

$$u = -V_h\sin(\Theta) \tag{10}$$

$$v = -V_h\cos(\Theta) \tag{11}$$

Here we consider the meteorological convention for the wind direction in which the wind direction is expressed in terms of the direction from which the wind originates (e.g., a northerly wind blows from the north to the south). In the climate forecast (CF) convention this parameter is denoted as *wind from direction* instead of *wind direction* to avoid ambiguity. Accordingly,

due to the applied convention for wind related parameters(i.e., $u$, $v$, $w$ and $\Theta$) we have a negative sign in front of $V_h$ in Equation 10 and 11. These expressions can be introduced in Equation 9, which yields a relation between $V_{radial}$, $V_h$ and $\Theta$:

$$V_{radial} = -V_h\cos(\varphi)(\sin(\Theta)\sin(\theta) + \cos(\Theta)\cos(\theta)) \tag{12}$$

Now lets consider that we have wind field which only exhibits a change of the horizontal wind speed amplitude with the height above ground level, and that this change can be described using the power law wind profile:

$$V_h(H) = V_{h_{ref}}\left(\frac{H}{H_{ref}}\right)^{\alpha} \tag{13}$$



where $V_{h_{ref}}$ is a reference/known horizontal wind speed at the reference/known height $H_{ref}$, $\alpha$ is a wind shear exponent, and $H$ is a height at which we are deriving the horizontal wind speed $V_h(H)$. It is important to remember that $H_{ref}$ and $H$ are expressed as the height above ground level.

5      We intend to probe this wind field with a scanning lidar at a certain point $M$ which is positioned at a certain height above ground level $H$ (Figure 2). To do this we would need to steer the laser beam towards the point of interest with a certain elevation and azimuth angle $\theta$ and $\varphi$ and acquire the backscatter signal at a range $R$ along the beam direction. The height $H$ can be expressed in terms of the height of ground below point $M$ above sea level $H_g$, height of lidar above sea level $H_l$, elevation angle $\varphi$ and range $R$:

$$H = R sin(\varphi) + (H_l - H_g) \tag{14}$$

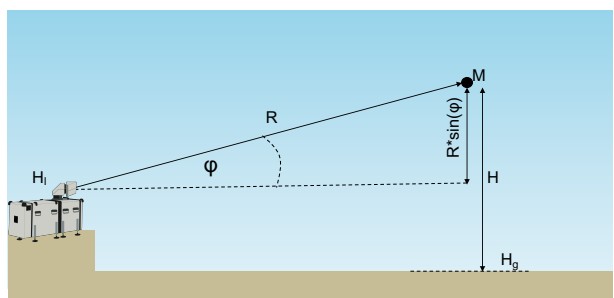

**Figure 2.** Single scanning lidar probing atmosphere: $M$ - probed measurement point, $H$ - height of $M$ above ground level, $H_g$ - height of ground below point $M$ above sea level, $H_l$ - height of lidar above sea level, $\varphi$ - elevation angle and $R$ - range

10      Therefore, Equation 13 can be modified such that it indicates scanning lidar parameters:

$$V_h(R, \varphi) = V_{h_{ref}} \left( \frac{R sin(\varphi) + (H_l - H_g)}{H_{ref}} \right)^\alpha \tag{15}$$

We can introduce Equation 15 in Equation 12 which yields:

$$V_{radial} = -V_{h_{ref}} \cos(\varphi) \cos(\theta - \Theta) \left( \frac{R sin(\varphi) + (H_l - H_g)}{H_{ref}} \right)^\alpha \tag{16}$$

With the last expression, we have all the required relation between $V_{radial}$, lidar parameters for probing of the atmosphere 15   $(R, \theta, \varphi)$ and wind field $(V_h, \Theta, \alpha)$, which we will use to derive a radial wind speed uncertainty model.

     The uncertainty model of $V_{radial}$ can be derived by applying the law of the uncertainty propagation while considering that the uncertainty contributions of $V_{radial}$ are independent from each other :

$$u_{V_{radial}}^2 = u_{est}^2 + \left( u_\varphi \frac{\partial V_{radial}}{\partial \varphi} \right)^2 + \left( u_\theta \frac{\partial V_{radial}}{\partial \theta} \right)^2 + \left( u_R \frac{\partial V_{radial}}{\partial R} \right)^2 \tag{17}$$





where $u_{V_{radial}}$, $u_{est}$, $u_\theta$, $u_\varphi$ and $u_R$ are radial wind speed, LOS estimation, azimuth, elevation and ranging uncertainties. These uncertainties are derived using dedicated calibration procedures which will be the topic of the follow-up publication. We relate to these uncertainties as to the intrinsic uncertainties of wind lidars.

Since we established the relation between the probing parameters and probed atmosphere we can derive partial derivatives from Equation 17:

$$\frac{\partial V_{radial}}{\partial \varphi} = -\frac{\alpha}{H_{ref}} R \cos(\theta - \Theta) \cos(\varphi)^2 V_{h_{\text{ref}}} \left( \frac{R\sin(\varphi) + (H_l - H_g)}{H_{ref}} \right)^{\alpha - 1} + \cos(\theta - \Theta) \sin(\varphi) V_{h_{\text{ref}}} \left( \frac{R\sin(\varphi) + (H_l - H_g)}{H_{ref}} \right)^\alpha$$
(18)

$$\frac{\partial V_{radial}}{\partial \theta} = \sin(\theta - \Theta) \cos(\varphi) V_{h_{\text{ref}}} \left( \frac{R\sin(\varphi) + (H_l - H_g)}{H_{ref}} \right)^\alpha$$
(19)

$$\frac{\partial V_{radial}}{\partial R} = -\frac{\alpha}{H_{ref}} \cos(\theta - \Theta) \cos(\varphi) \sin(\varphi) V_{h_{\text{ref}}} \left( \frac{R\sin(\varphi) + (H_l - H_g)}{H_{ref}} \right)^{\alpha - 1}$$
(20)

Let's examine the derived relations, especially Equation 18. This partial derivative contains two terms. The first term represents the translation of the elevation uncertainty to the height uncertainty. The second term represents the uncertainty due to the 'wrong' projection of the wind vector to the LOS. The second term tends to be much smaller than the first term, since $\sin(\varphi)$ is close to zero for low elevation angles, while $\cos(\varphi)$ is close to one. Also, as expected the first term grows as the range becomes higher since in a combination with the elevation angle uncertainty the increase in range leads to the increase in the height uncertainty. As we can see both terms contain $\cos(\theta - \Theta)$ which means that when the beam is aligned with the wind direction the partial derivative reaches maximum value. Otherwise, when the beam is perpendicular to the wind this partial derivative is equal to zero. Since in our wind field model the wind only changes with height the second partial derivative (Equation 19) reflects the uncertainty due to the 'wrong' projection of the wind vector to the LOS. It is important to notice that contrary to the previous partial derivative this partial derivative reaches a maximum value when the beam is perpendicular to the wind. The rationale behind this is that the projection of the wind vector to LOS is the most sensitive to the uncertainty when the beam is perpendicular to the wind. As the range uncertainty translates to the height uncertainty, the associated partial derivative (Equation 20) tends to maximum when the wind and the beam direction are aligned. However, for the low elevation angles the range uncertainty translation into the height uncertainty is small ($\sin(\varphi)R$), on the other hand even though $\sin(\varphi)R$ is maximum for $90°$ due to the selected wind field model the projection of the wind to the LOS is zero. Therefore, the partial derivative in Equation 20) reaches a maximum value when the wind and the beam directions are aligned and when $\varphi$ is $45°$.

Since we produced the partial derivatives we can re-introduce $H$ back in Equation 18 - 20 for the sake of simplicity:

$$\frac{\partial V_{radial}}{\partial \varphi} = -\frac{\alpha}{H_{ref}} R \cos(\theta - \Theta) \cos(\varphi)^2 V_{h_{\text{ref}}} \left( \frac{H}{H_{ref}} \right)^{\alpha - 1} + \cos(\theta - \Theta) \sin(\varphi) V_{h_{\text{ref}}} \left( \frac{H}{H_{ref}} \right)^\alpha$$
(21)



$$\frac{\partial V_{radial}}{\partial \theta} = \sin(\theta - \Theta)\cos(\varphi)V_{h_{\text{ref}}}\left(\frac{H}{H_{ref}}\right)^{\alpha} \tag{22}$$

$$\frac{\partial V_{radial}}{\partial R} = -\frac{\alpha}{H_{ref}}\cos(\theta - \Theta)\cos(\varphi)\sin(\varphi)V_{h_{\text{ref}}}\left(\frac{H}{H_{ref}}\right)^{\alpha-1} \tag{23}$$

To derive the combined radial velocity uncertainty model we only need to introduce expressions for partial derivatives given in Equation 18 - 20 into Equation 17. We omit displaying the derived model due to required space.

## 5   3.3   Dual-Doppler wind speed uncertainty model

Let's now consider a dual-Doppler setup in which we have two scanning lidars directing the beams to meet at the same point $M$ in the atmosphere sensing two independent radial velocities $V_{radial_1}$ and $V_{radial_2}$ (Figure 3). Neglecting the vertical wind speed we can write the following relation between the radial velocities and the probed horizontal wind speed components $u$ and $v$:

$$10 \quad V_{radial_1} = u\sin(\theta_1)\cos(\varphi_1) + v\cos(\theta_1)\cos(\varphi_1) \tag{24}$$

$$V_{radial_2} = u\sin(\theta_2)\cos(\varphi_2) + v\cos(\theta_2)\cos(\varphi_2) \tag{25}$$

where $\theta_1$ and $\varphi_1$ are the azimuth and elevation angles of the first lidar, while $\theta_2$ and $\varphi_2$ are the azimuth and elevation angles of the second lidar.

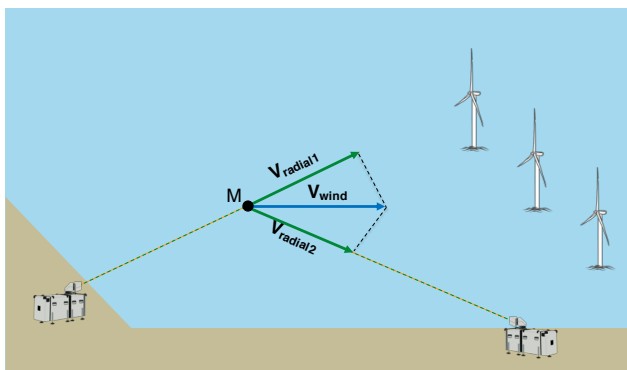

**Figure 3.** Dual-Doppler system probing atmosphere: $M$ - probed measurement point, $V_{radial_1}$ - radial wind speed acquired by first lidar, $V_{radial_2}$ - radial wind speed acquired by second lidar, and $\boldsymbol{V}_{\text{wind}}$ - wind vector.





Taking into account the above given set of equations we can expressed $u$ and $v$ in terms of $V_{radial_1}$ and $V_{radial_2}$:

$$u = \frac{V_{radial_1}\cos(\theta_2)\cos(\varphi_2) - V_{radial_2}\cos(\theta_1)\cos(\varphi_1)}{\cos(\varphi_1)\cos(\varphi_2)\sin(\theta_1 - \theta_2)} \tag{26}$$

$$v = \frac{V_{radial_2}\sin(\theta_1)\cos(\varphi_1) - V_{radial_1}\sin(\theta_2)\cos(\varphi_2)}{\cos(\varphi_1)\cos(\varphi_2)\sin(\theta_1 - \theta_2)} \tag{27}$$

For the sake of calculating the dual-Doppler uncertainty of the retrieved horizontal wind speed components, we will simplify
these expressions by assuming that all the elevation angles are small and that $\cos(\varphi_1)$ and $\cos(\varphi_2)$ are approximately equal
to 1. Note that the effect of elevation (and azimuth) angle uncertainty is still retained in the expression for the radial speed
uncertainty (e.g., Equation 17) that we will later use in the dual-Doppler uncertainty derivation.

Simplifying for the zero elevation gives:

$$u = \frac{V_{radial_1}\cos(\theta_2) - V_{radial_2}\cos(\theta_1)}{\sin(\theta_1 - \theta_2)} \tag{28}$$

$$v = \frac{V_{radial_2}\sin(\theta_1) - V_{radial_1}\sin(\theta_2)}{\sin(\theta_1 - \theta_2)} \tag{29}$$

Finding the root of the squared sum of $u$ and $v$ yields the horizontal wind speed:

$$V_h = \frac{\sqrt{V_{radial_1}^2 - 2V_{radial_1}V_{radial_2}\cos(\theta_1 - \theta_2) + V_{radial_2}^2}}{\sin(\theta_1 - \theta_2)} \tag{30}$$

The uncertainty of the horizontal wind speed $U_{V_h}$ can be derived by applying the propagation of uncertainty, thus calculating
partial derivatives of $V_h$ for $V_{radial_1}$ and $V_{radial_2}$:

$$U_{V_h} = \sqrt{\left(\frac{\partial V_h}{\partial V_{radial_1}}u_{V_{radial_1}}\right)^2 + \left(\frac{\partial V_h}{\partial V_{radial_2}}u_{V_{radial_2}}\right)^2} \tag{31}$$

It must be noted that we don't perform the partial derivative of $V_h$ for $\theta$ and $\varphi$ because the pointing uncertainty is already
included when calculating $U_{radial_1}$ and $U_{radial_2}$.

The partial derivatives in Equation 31 are evaluated as:

$$\frac{\partial V_h}{\partial V_{radial_1}} = \frac{1}{\sin(\theta_1 - \theta_2)}\frac{V_{radial_1} - V_{radial_2}\cos(\theta_1 - \theta_2)}{\sqrt{V_{radial_1}^2 + V_{radial_2}^2 - 2V_{radial_1}V_{radial_2}\cos(\theta_1 - \theta_2)}} \tag{32}$$

and,

$$\frac{\partial V_h}{\partial V_{radial_2}} = \frac{1}{\sin(\theta_1 - \theta_2)}\frac{V_{radial_2} - V_{radial_1}\cos(\theta_1 - \theta_2)}{\sqrt{V_{radial_1}^2 + V_{radial_2}^2 - 2V_{radial_1}V_{radial_2}\cos(\theta_1 - \theta_2)}} \tag{33}$$





Equation 32 and 33 can be simplified taking into account Equation 30:

$$\frac{\partial V_h}{\partial V_{radial_1}} = \frac{V_{radial_1} - V_{radial_2}\cos(\theta_1 - \theta_2)}{V_h\sin(\theta_1 - \theta_2)^2} \tag{34}$$

and,

$$\frac{\partial V_h}{\partial V_{radial_2}} = \frac{V_{radial_2} - V_{radial_1}\cos(\theta_1 - \theta_2)}{V_h\sin(\theta_1 - \theta_2)^2} \tag{35}$$

5    Substituting Equation 34 and 35 in Equation 31 gives the expression for the dual-Doppler uncertainty for the horizontal wind speed:

$$U_{V_h} = \frac{1}{V_h\sin(\theta_1 - \theta_2)^2} * \sqrt{(V_{radial_1} - V_{radial_2}\cos(\theta_1 - \theta_2))^2 u_{V_{radial_1}}^2 + (V_{radial_2} - V_{radial_1}\cos(\theta_1 - \theta_2))^2 u_{V_{radial_2}}^2} \tag{36}$$

It is important to notice that similar to derivations in Davies-Jones (1979) and Stawiarski et al. (2013) Equation 36 contains the term $\frac{1}{V_h\sin(\theta_1-\theta_2)^2}$ which acts as an amplifier of total uncertainty since as the between-beam angle $(\theta_1 - \theta_2)$ tends to zero

10    the whole term tends to infinity.

### 3.4   Dual-Doppler wind direction uncertainty model

With the same approach as in case of horizontal wind speed uncertainty we can derive the dual-Doppler uncertainty of the wind direction. We start first with a mathematical expression for the wind direction:

$$\Theta = \arctan\left(\frac{v}{u}\right) \tag{37}$$

15    The uncertainty of the wind direction $\Theta$ is:

$$U_\Theta = \sqrt{\left(\frac{\partial\Theta}{\partial V_{radial_1}}u_{V_{radial_1}}\right)^2 + \left(\frac{\partial\Theta}{\partial V_{radial_2}}u_{V_{radial_2}}\right)^2} \tag{38}$$

The partial derivatives in Equation 38 are evaluated as:

$$\frac{\partial\Theta}{\partial V_{radial_1}} = \frac{-V_{radial_2}\sin(\theta_1 - \theta_2)}{V_{radial_1}^2 - 2V_{radial_2}V_{radial_1}\cos(\theta_1 - \theta_2) + V_{radial_2}^2} \tag{39}$$

and

$$\frac{\partial\Theta}{\partial V_{radial_2}} = \frac{V_{radial_1}\sin(\theta_1 - \theta_2)}{V_{radial_1}^2 - 2V_{radial_2}V_{radial_1}\cos(\theta_1 - \theta_2) + V_{radial_2}^2} \tag{40}$$

which if we substitute them in Equation 38 yields the uncertainty model:

$$U_\Theta = \sqrt{\frac{(u_{V_{radial_1}}^2 V_{radial_2}^2 + u_{V_{radial_2}}^2 V_{radial_1}^2)\sin(\theta_1 - \theta_2)^2}{(V_{radial_1}^2 - 2V_{radial_2}V_{radial_1}\cos(\theta_1 - \theta_2) + V_{radial_2}^2)^2}} \tag{41}$$





This equation can be further simplified:

$$U_\Theta = \frac{1}{V_h^2 |\sin(\theta_1 - \theta_2)|} \sqrt{(u_{V_{radial_1}}^2 V_{radial_2}^2 + u_{V_{radial_2}}^2 V_{radial_1}^2)} \tag{42}$$

Once again the between-beam angle $(\theta_1 - \theta_2)$ will act as an amplifier of the uncertainty.

### 3.5 Digitalizing the uncertainty models

The previously described uncertainty models have been digitalized using Python and a set of public Python libraries resulting in a Python package *YADDUM* (Vasiljevic, 2019). The package has been made public (open source), it is versioned on Github and persisted using Zenodo (Vasiljevic, 2019). At the time of writing this manuscript, version 0.1.3 has been released. Using *YADDUM* end-users can assess the uncertainty of dual-Doppler retrievals of wind speed and wind direction. We will demonstrate *YADDUM* in the following section.

## 4 Demonstration of uncertainty models

To demonstrate our speed and direction uncertainty models, we perform an uncertainty analysis of the RUNE field campaign (Floors et al., 2016). The RUNE campaign together with other instrumentation, included a dual-Doppler setup comprising two long-range WindScanners (Vasiljevic et al., 2016). These two scanning lidars are denoted as *koshava* and *sterenn* in Table 1 of Floors et al. (2016). We will assess how accurate this dual-Doppler setup can acquire wind speed and wind direction over a large area. For the dual-Doppler setup, we will calculate so-called uncertainty maps that graphically depict modeled uncertainties. Specifically, we will derive radial wind speed, horizontal wind speed and wind direction uncertainty maps. We will also derive maps that will show how each intrinsic uncertainty term contributes to the radial uncertainty. To do this we will use *YADDUM*, which as mentioned earlier, already implements the speed and direction uncertainty models derived in this paper.

To simplify demonstration, we will assume that the two lidars are positioned at sea level (i.e., at 0 m asl) and that the surrounding area is flat terrain with a height of 0 m above sea level. Accordingly, the positions of *koshava* and *sterenn* in Easting, Northing, and height above sea level are 446080.03 m, 6259660.30 m and 0 m respectively, and 445823.66 m, 6263507.90 m and 0 m respectively.

*YADDUM* is parametrized with the lidar positions and values of their intrinsic uncertainties, followed by the atmospheric model (power-law profile) and positions of the measurement points. For measurement points, we will use a horizontal mesh at 100 m above sea level (the elevation angle adjusts to give this height at each measurement point) with a resolution of 10 m extending 5000 m in Easting and Northing from the barycenter between the two lidars. We will perform two runs of *YADDUM*. In the first run, the wind direction will be 0° (wind coming from the North), in the second run the wind direction will be set to 270°. The parametrization of *YADDUM* is provided in Table 1. The uncertainties in the table are standard uncertainties (i.e., the coverage factor $k$ is equal to 1) with values we typically find during a lidar calibration. Our results are also all presented as standard uncertainties.





**Table 1.** *YADDUM* parametrization

| Parameters | Value |
| --- | --- |
| Mesh extent | 5000 m |
| Mesh resolution | 10 m |
| Wind speed $V_{H_{ref}}$ | 10 m/s |
| Reference height $H_{ref}$ | 100 m agl |
| Upward air velocity | 0 m/s |
| Shear exponent $\alpha$ | 0.2 |
| Wind direction $\Theta$ | $1^{st}$ run : 0°, $2^{nd}$ run: 270° |
| Estimation uncertainty $u_{est}$ | 0.1 m/s |
| Azimuth uncertainty $u_{azimuth}$ | 0.1° |
| Elevation uncertainty $u_{elevation}$ | 0.1° |
| Range uncertainty $u_{range}$ | 10 m |

The results of the two *YADDUM* runs are shown in Figures 4 - 7. Let's first inspect the radial velocity uncertainties and associated uncertainty contributors. Looking at the results of the first run (wind direction equal to 0°) shown in Figure 4 we can see that the radial velocity (top row of plots) increases if we go from the lidar position towards North or South (i.e., aligned with the wind direction). If we go East or West from the lidar the radial uncertainty is constant and equal to $u_{est}$ ($\sim$ 0.1 m/s).

The reason for this is that the elevation uncertainty is the main contributor to the radial uncertainty apart from the estimation uncertainty. This is evident from the remaining rows of plots shown in Figure 4 if we look at the range of values each contributor has (see the color bars). As discussed in Section 4.2 the elevation contribution attains the maximum value when the laser beam direction is aligned with the wind direction. If we zoom into the area around the lidar (right plot in the second row from top) we can see that elevation contribution very close to the lidar reaches high value, then tends to zero, switch the sign and keeps

on increasing. If we look at Equation 21 we can see that the two terms have different signs and that the second term will reach the maximum value when the elevation angle is 90°. Opposite to the elevation contribution, the azimuth contribution achieves the maximum value when the beam is perpendicular to the flow (see Equation 22). In the case of the range contribution (see Equation 23), the maximum value is achieved when the beam is aligned with the flow and when the elevation angle is 45°. This can be seen from the bottom right plot in Figure 4, where the maximum values are located 100 m towards North and South

from the lidar location. Figure 5 shows the results of the second run (wind direction equal to 270°). Evidently, the results are rotated by 90° if we compare Figure 4 and Figure 5.

When assessing wind speed and wind direction uncertainties it is important to remember that in addition to the radial wind speed uncertainty we have the amplification factor due to the between-beam angle. Therefore, as the two beams become parallel to each other we should expect to see an increase in the wind speed and wind direction uncertainties. By analyzing

Figure 6 and Figure 7 to certain extent we do encounter expected behaviour. However, we can notice that the amplification of



**Figure 4.** Uncertainty maps for *koshava* for wind direction of $0°$: left plots show entire mesh area while right images show area zoomed around *koshava*. Top row plots show radial uncertainty map, plots in second row from top show contribution of elevation uncertainty to the radial uncertainty, plots in third row from top show contribution of azimuth uncertainty to the radial uncertainty, and plots in bottom row show contribution of range uncertainty to the radial uncertainty.



**Figure 5.** Uncertainty maps for $koshava$ for wind direction of $270°$: left plots show entire mesh area while right images show area zoomed around $koshava$. Top row plots show radial uncertainty map, plots in second row from top show contribution of elevation uncertainty to the radial uncertainty, plots in third row from top show contribution of azimuth uncertainty to the radial uncertainty, and plots in bottom row show contribution of range uncertainty to the radial uncertainty.





the uncertainties strongly depends on wind direction and that the wind speed and wind direction uncertainties have somewhat contrasting behavior to each other. Specifically, when the wind is coming from the North the wind direction uncertainty is more affected than the wind speed uncertainty in the area of the mesh where the between-beam angle tends to zero. On the other hand, when the wind is coming from the West it is the wind speed uncertainty which is more affected in the same area.

To better understand these results it is useful to analyze values of the denominators and numerators of Equation 36 and Equation 42 which are depicted in the mid and bottom rows of Figure 6 and Figure 7. It is important to remember that the denominators are independent of the wind direction, thus the mid row plots in Figure 6 are identical to each other, which is the same situation in Figure 7.

For both the wind speed and wind direction uncertainty, the denominator tends to have values close or equal to zero in the
area around the line which connects the two lidars. This is more obvious for the wind speed uncertainty (the area marked with yellow in the mid-row plots in Figure 6) since the denominator in Equation 36 includes $\sin(\theta_1 - \theta_2)^2$, while the denominator in Equation 42 contains $\sin(\theta_1 - \theta_2)$. Either way, the area around the line which connects the two lidars should experience a significant amplification of the numerator values. But, the area where the denominator tends to have low values corresponds to the area where the numerator has also low values.

However, the numerator values are dependent on the wind direction. Considering the wind speed uncertainty, for the North wind the overlap between the area where the numerator and denominator have low values is greater than when the wind is coming from the West (Figure 6). This is the opposite in the case of wind direction uncertainty (Figure 7).

For a designer of dual-Doppler field campaigns, probably more relevant information would be to know a minimum between-beam angle which will secure accurate wind speed and direction retrievals. Figure 8 shows histograms of wind speed and
wind direction uncertainties for the two wind directions and three different limits for the between-beam angle. Table 2 and 3 summarize the histogram results. Based on these results maintaining the between-beam angle larger than 30° during the measurements will secure accurate wind speed and direction retrievals. A similar value was suggested by Davies-Jones (1979) in case of dual-radar systems.

**Table 2.** Histogram results for wind speed uncertainty

| Parameters | $\Theta = 0°$ | | | $\Theta = 270°$ | | |
|---|---|---|---|---|---|---|
| | $(\theta_1 - \theta_2) >$ | | | | | |
| | 20° | 30° | 40° | 20° | 30° | 40° |
| Mean [m/s] | 0.28 | 0.23 | 0.20 | 0.22 | 0.17 | 0.14 |
| Minimum [m/s] | 0.09 | 0.09 | 0.09 | 0.01 | 0.01 | 0.01 |
| Maximum [m/s] | 1.24 | 0.63 | 0.33 | 0.53 | 0.34 | 0.22 |
| Standard deviation [m/s] | 0.12 | 0.08 | 0.05 | 0.10 | 0.05 | 0.03 |



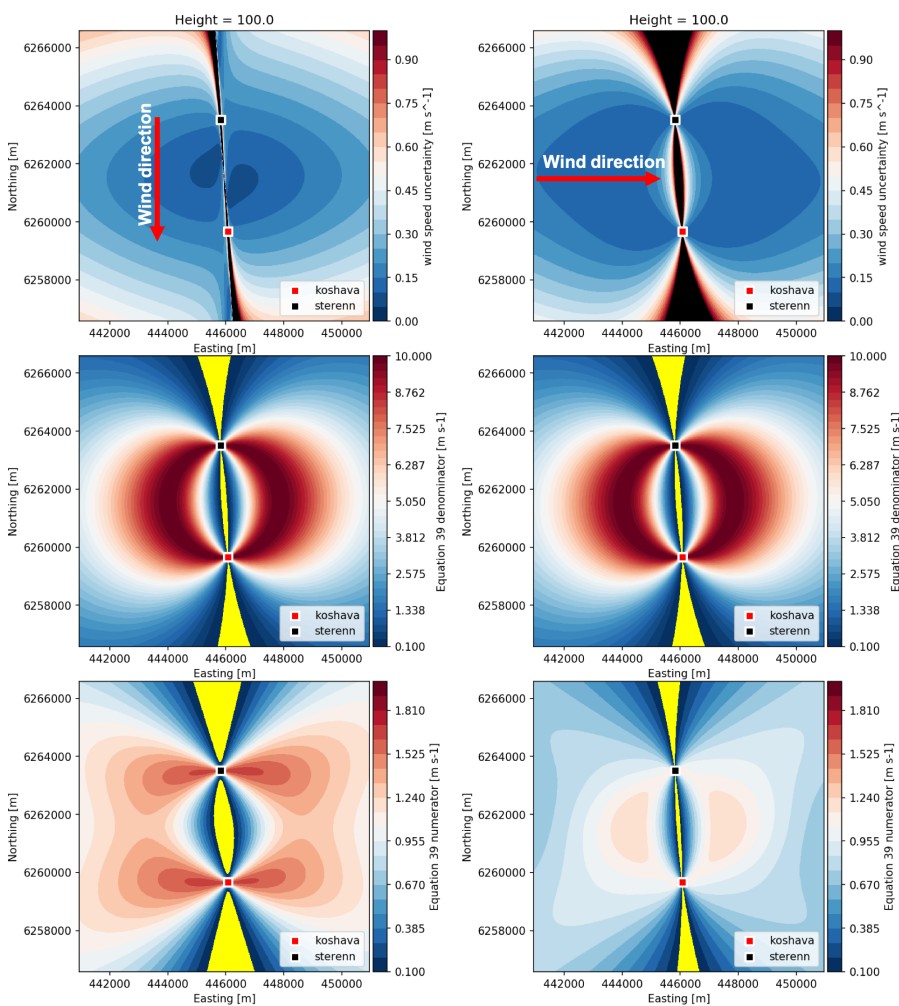

**Figure 6.** Dual-Doppler uncertainty maps for wind speed: left column plots - results for wind direction of $0°$, right column plots - results for wind direction of $270°$, top row plots - total wind speed uncertainty, mid row plots - Equation 36 denominator values, bottom row plots - Equation 36 nominator values. The plot areas colored in yellow indicate that the values go below 0.1 m/s.



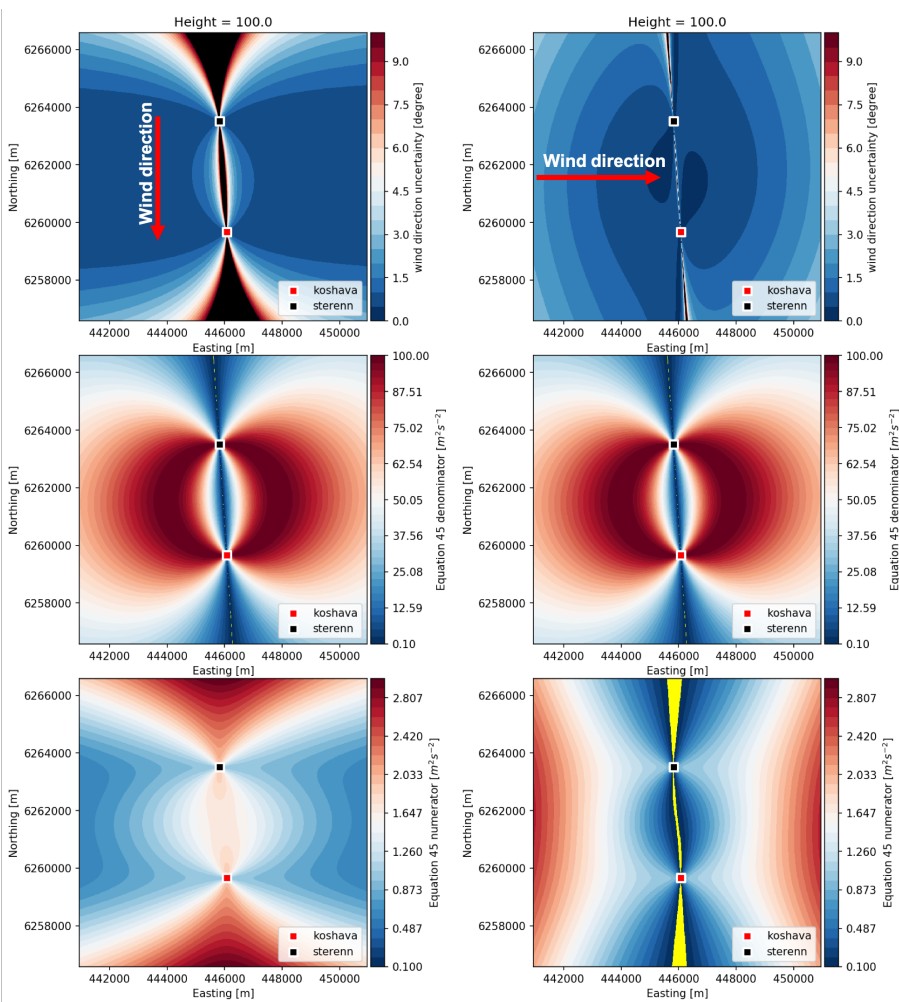

**Figure 7.** Dual-Doppler uncertainty maps for wind direction: left column plots - results for wind direction of $0°$, right column plots - results for wind direction of $270°$, top row plots - total wind direction uncertainty, mid row plots - Equation 41 denominator values, bottom row plots - Equation 41 nominator values. The plot areas colored in yellow indicate that the values go below 0.1 $(m/s)^2$



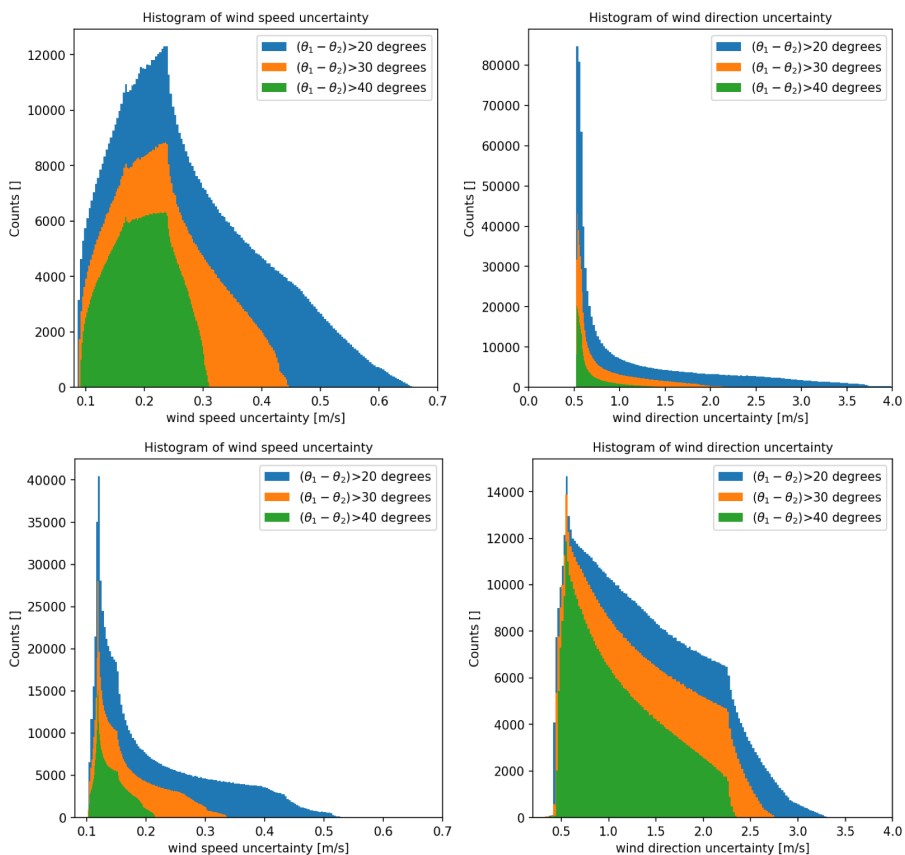

**Figure 8.** Histograms of wind speed and wind direction uncertainties: top row plots - results for wind direction of $0°$, bottom row plots - results for wind direction of $270°$, left column plots - wind speed uncertainty histogram, right column plots - wind direction uncertainty histogram.

**Table 3.** Histogram results for wind direction uncertainty

| Parameters | $\Theta = 0°$ | | | $\Theta = 270°$ | | |
|---|---|---|---|---|---|---|
| | $(\theta_1 - \theta_2) >$ | | | | | |
| | $20°$ | $30°$ | $40°$ | $20°$ | $30°$ | $40°$ |
| Mean [°] | 1.22 | 0.84 | 0.68 | 1.38 | 1.27 | 1.13 |
| Minimum [°] | 0.52 | 0.52 | 0.52 | 0.07 | 0.07 | 0.07 |
| Maximum [°] | 4.26 | 2.22 | 1.51 | 3.32 | 2.77 | 2.35 |
| Standard deviation [°] | 0.83 | 0.37 | 0.19 | 0.64 | 0.57 | 0.49 |





## 5 Discussion

### 5.1 Novelty and value of the model

Our model is unique in including the real-life uncertainties associated with a scanning lidar - LOS speed uncertainty as well as angular and range uncertainties. To our knowledge other models only consider LOS speed noise and do not correctly include the

effect of elevation angle errors and range errors on the actual measurement height. Here we have a model that for the first time can be used, for example in the domain of wind energy, to provide estimates of wind speed and wind direction uncertainties in both wind resource and power performance validation applications.

A second novelty is the availability of the model in a ready-coded Python package. Hopefully this will foster its use and also simplify benchmarking with other approaches. The digitisation also lends itself to optimisation exercises so that future

campaigns can benefit from an uncertainty optimised experimental design.

### 5.2 Limitations of the model

It is always vitally important to be aware of the limitations that we have built in during the derivation. Let us re-cap and re-examine our assumptions.

Firstly we assumed that our lidar is a point measurement device - ignoring that the wind speed is actually gathered as a

weighted average over a length of the line of sight - this process usually encapsulated in a probe length parameter. Where there are strong local gradients such that a weighted average is no longer a valid representation of the wind speed at the nominal point, extra uncertainties will be introduced and should then be added to the model. For example measuring in or close to a wind turbine wake could introduce such errors. Conversely, in many applications the local wind gradients will be small and will not introduce errors of this type.

Two other important and related assumptions are that elevation angles are low and (therefore) that vertical wind speed components can be ignored. Whilst the low elevation angle restriction allows us to ignore vertical components it also of course restricts us to applications where this is fulfilled. Later in our derivation we also assumed small elevation angles to simplify our expressions. Thus even in cases with essentially horizontal flow we are not entitled to use our model where the elevation angle exceeds say $10°$. We are probably more justified in using the model where the elevation angles remain low but where there are

significant vertical components since these will have only very contributions to the radial speeds.

A power-law wind speed profile was also assumed. Whilst it is recognised that this is an unrealistic model for a full wind profile it does provide us with a convenient (easy to differentiate) model with a readily recognisable (and single) parameter - the shear coefficient $\alpha$. As long as a reasonable value for $\alpha$ is chosen (valid at the measurement height) the model will provide a reasonable estimate of the uncertainties that arise from incorrect measurement height - an important feature of our model.

Whilst vertical profiles were allowed, we have not included any horizontal gradients. Here the arguments pertaining to point versus finite probe length also apply. Only very severe horizontal gradient (significant changes over the probe length) will cause any meaningful errors.



Our uncertainty model has also assumed that there are no correlations between any of the uncertainties. With two lidars, possibly of the same type, this may not be completely true. It is not unreasonable to assume that for example, a line of sight estimation error is common to both lidars (either through common design or common calibration). Such a correlation would tend to slightly increase the combined uncertainty for the u component (where the radial speeds add) and reduce it for the v

component (where the radial speeds subtract).

Whilst we have not explicitly included noise term for the LOS speed estimation uncertainty, such a term can be considered as implicitly contained in the value. The LOS uncertainty comprises both a random (noise) and a systematic part. Usually the latter will dominate and is largely determined by the uncertainty of the reference speed used in the calibration (often a cup anemometer).

Finally, one can argue against the propagation of uncertainties method itself that it easily becomes very complicated when the input model is expanded. Instead, for more complicated models a Monte Carlo simulation type approach can be more suitable. With a Monte Carlo simulation the measurement process is mimicked by calculating the measurement output for a number of randomly chosen input parameters but with known statistics corresponding to their expectation value and uncertainty. In this way the simulation can in certain cases automatically include some intricate correlations that can otherwise be difficult

to quantify. The latter is for example true when calculating the horizontal wind speed uncertainty based on the orthogonal wind speed components $u$ and $v$, but is not an issue for our model where the uncertainty is found directly from the radial velocities and associated uncertainties. Besides being mathematically less complex and simpler to implement, Monte Carlo simulations furthermore have the advantage of enabling integration of other (perhaps more realistic) flow models (see *mocalum* in Vasiljevic, 2020).

**6 Conclusions**

Uncertainty models for wind speed and wind direction have been developed for a pair of scanning lidars operating in dual-Doppler configuration. The models are developed analytically and therefore some assumptions have been necessary to keep the algebra manageable. Most importantly, the lidars are assumed to operate with low elevation angles and away from areas of extreme horizontal or vertical shear. A power-law model has been used to model the vertical wind profile and this allows

the important effects of elevation angle and range uncertainty to be reasonably estimated. A digitalization of the models, YADDUM, is available as an open source Python package.

The models have been demonstrated using a previous field experiment. It can be seen how the speed and direction uncertainties are combinations of the radial wind speed uncertainties amplified by the dual-Doppler reconstruction algorithm. The analytical development provides considerable insight into to observed patterns in the uncertainty maps for the demonstration

experiment. Future dual-Doppler measurement campaigns operating within the domain of the discussed limitations, can use the YADDUM model and package directly to both optimise the experimental design and estimate the uncertainties on the obtained results.





*Code availability.* YADDUM is available on GitHub https://github.com/niva83/YADDUM

*Author contributions.* N.V. and M.C. developed mathematical model for dual-Doppler uncertainty. N.V. created YADDUM based on the model. A.P. validated model. N.V. wrote a first draft of paper. M.C. and A.P. reviewed and improved paper.

*Competing interests.* The authors declare no conflict of interest

5  *Acknowledgements.* The financial support for the presented work has been provided by the RUNE project, which is funded by the ForskEL program (12263), and the RECAST project, which is funded by Innovation Fund Denmark (7046-00021B).



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
