# Peer review of "Uncertainty model for dual-Doppler retrievals of wind speed and wind direction"

_Atmospheric Measurement Techniques, 2020_

## Referee Comment (RC1) · Anonymous Referee #1 · 7 Oct 2020

Review of "Uncertainty model for dual-Doppler retrievals of wind speed and wind direction" by Nikola Vasiljevi et al.

General comments

This paper presents an analytical model for the uncertainty in horizontal wind speed and direction estimates obtained using a dual-Doppler sampling technique. The model attempts to account for ranging and pointing errors using a very simplistic atmospheric model, i.e. one in which the height variation is modeled using a power-law and the flow is assumed to be horizontally homogeneous. As a result, the radial velocity uncertainty contains no azimuthal dependence, which is not realistic. The author's don't provide a compelling justification for this assumption. I believe the problem calls for a more stochastic approach in which 3D variations are considered, as the authors suggest in the "discussion" section of this paper. Another limitation is that the error in the observed radial velocity is treated as a constant, when in fact it varies strongly with range (and depends on aerosol loading) in real lidar systems. At longer ranges I would imagine that this is probably the dominate source of random error in most dual-Doppler lidar set ups. The only practical utility I see is that this model may be useful is as a planning tool for dual-Doppler deployments (provided the range-dependence of the radial velocity uncertainty is accounted for). In my view, this paper is incomplete, in that there is more that should be done to make this a useful contribution. I recommend against publication in its current form.

Specific comments

Abstract: The abstract is short on results and it does not adequately convey important assumptions, i.e. that the elevation angle must be small, and the wind field is assumed to be horizontally homogeneous.

page 1 line 5: change "…available though as …" to "…available as …"

page 4 lines 5-10: Here the author critiques the inconsistent use of terminology (i.e. precision, accuracy, trueness??) but does not offer to clarify the meanings of these various terms.

Equation 4: Change "$V_{rad1}$" to "$V_{radial1}$"

Page 5, line 2: The author states "Due to the above-mentioned amplification of uncertainties caused by the Dual-Doppler setup…" But the author has not explained the "amplification" parameter yet at this point in the paper.

Page 5, line 7: Here and many other places in the paper the author refers to the radial velocity as the "radial wind speed." Wind speed is defined as the magnitude of the velocity vector. So "radial wind speed" doesn't make any sense. Please change all occurances of "radial wind speed" to "radial velocity."

Page 5, lines 7 and 8: The author poses two questions about accuracy, without ever defining "accuracy" (or for that matter without defining uncertainty, error, precision, or trueness???).

Page 5, line 12: The author introduces the beam direction parameter "D", but it is never used anywhere else in the paper. Please delete.

Equation 5: This equation gives the true radial velocity as a function of the measured radial velocity, This cant be right, because it implies you can somehow correct for random errors. This equation would make more sense if the right-hand-side was the radial velocity uncertainty. Please clarify.

Page 6, lines 4: Here the author discusses the atmospheric flow model. Winds are assumed to be horizontally homogeneous and the wind speed profile is modeled using a power-law relationship. As I point out above, this is an oversimplification. As a result, the uncertainty model would not be useful for assigning uncertainty estimates to actual measurements performed in the field (what value of alpha to use?). The only utility I can see is as a field campaign planning tool that provides crude estimate of expected errors for a given dual-Doppler set up (provided the range-dependence of $u_{est}$ is included). For the results of this study to be of practical use a more realistic model of the atmospheric flow is needed (e.g. Rod Frehlich's approach using the Von Karman spectrum). This would probably make any sort of analytical solution impossible.

Page 7, line 14: The standard meteorological definition of "wind direction" is the direction from which the wind blows. You could simply point this out, rather than introduce a non-standard term like "wind from direction".

Equation 17: $u_{est}$ is range dependent.

Page 9, line 2: When referring to uVradial , uest, u_, u' and uR the author states "These uncertainties are derived using dedicated calibration procedures which will be the topic of the follow-up publication." This is partly why I feel this study is not complete. These error estimates are central to this paper, and need to be discussed here, particularly $u_{est}$, which appears to be treated as a constant in this study, but should be range dependent.

Page 13, line 5: I don't believe "digitalized" (or digitize) is appropriate here. It would be more accurate to say that the model was programmed in Python.

Table 2 and 3: The top row (parameters) is confusing because its not clear how these relate to the numbers below. In the caption I would suggest that you explain that these numbers represent averages for theta=0 and theta=270.

Figure 8. The units for the wind direction uncertainty are shown as m/s. Should they be in degrees?. Please clarify.

Page 22, lines 3-5.The author states "Such a correlation would tend to slightly increase the combined uncertainty for the u component (where the radial speeds add) and reduce it for the v component (where the radial speeds subtract)." This is not generally true. The author seems to be referring to a specific dual-Doppler configuration.

Page 22, lines 7-9: The author states "The LOS uncertainty comprises both a random (noise) and a systematic part. Usually the latter will dominate and is largely determined by the uncertainty of the reference speed used in the calibration (often a cup anemometer)." The second sentence isn't generally true (that bias is larger than precision), unless the author is referring specifically to the behavior of their uncertainty model. Please clarify.

Page 22, lines 12-13: The author states "With a Monte Carlo simulation the measurement process is mimicked by calculating the measurement output for a number of randomly chosen input parameters but with known statistics corresponding to their expectation value and uncertainty." I concur with this approach.

---

## Referee Comment (RC2) · Anonymous Referee #2 · 8 Jan 2021

Review of "*Uncertainty model for dual-Doppler retrievals of wind speed and wind direction*" by Vasiljevic et al

**General Comments**

This paper aims to present an analytical model that accounts for pointing uncertainty in addition to the classical LOS uncertainty in dual-Doppler lidar configurations. There are numerous assumptions that must be made to solve the equations analytically. The most critical are that winds are horizontally homogeneous, winds change in height according to a power-law, uncertainty is uncorrelated, the lidars use shallow elevation angles, and vertical velocities are minimal. These assumptions are incredibly limiting. A more appropriate approach may be using Monte Carlo simulations, as suggested by the author in their discussion. In the current form, this reviewer feels that the developed model does not make a significant contribution to the field. Therefore, I do not recommend publication.

**Specific Comments (in no particular order):**

- The assumptions that must be applied to derive the theoretical uncertainty *severely* limit the variety of cases for which the algorithm can be applied. It is evident that there is a specific use case for this algorithm rather than it being a generalized model for dual-doppler setups. If so, this use case should be indicated (e.g. for land based lidar setups monitoring offshore wind farms as stated in the introduction)
- The uncertainty of the LOS velocity is treated as constant, which is not necessarily true. In reality, this is a function of range and aerosol backscatter. In turn, this also violates the assumption that the uncertainty contributors are uncorrelated.
- There should be a comparison between previously used models (i.e. only accounting for LOS uncertainty) and this model to prove that it does in fact perform better and/or show magnitude of differences between the two models
- Figure 4-7 should be modified to not use a diverging colorbar for non-diverging fields. This leads to misleading results and/or difficult interpretation. Continuous fields (i.e. wind speed uncertainty) should use continuous colorbars. This is well established in literature (e.g. Stauffer et al. 2015).
- Figure 4 and others: Can you please comment on how to interpret the negative uncertainty that your figures show? Does this indicate that there is less uncertainty in negative areas or should I be looking at the magnitude of the uncertainty? Please clarify. If the latter, please plot the magnitude.

- The abstract is quite short and devoid of any discussion of limitations of the algorithm and any results or takeaways from this study. Please include these topics
- Page 2 Lines 21-22: What is an acceptable amount of error for wind energy applications?
- Page 9 Line 2: I think this is an important piece that is missing from this paper. There is really no proof that the examples given have uncertainty magnitudes that you indicate. Additionally, anyone wanting to use this model can't unless they first have the uncertainties of their system.
- Figure 4 and others: It's generally good practice to normalize things when talking about relative contributions. I think this would communicate more clearly that the elevation uncertainty is the largest contributor.
- Page 21 Lines 26-28: I think it would be good to also show a sensitivity study on the alpha parameter.
- Page 22 Lines 10-19: I think this is really the way this problem should be addressed. The authors themselves state that Monte Carlo simulations are mathematically less complex, simpler to implement, and able to handle more realistic flows.

**Technical Comments (in no particular order; representative, not comprehensive):**

- There are many spots where in text citations should be parenthetical citations (e.g. Page 1 Line 11), and vice versa. Please check to make sure all citations are formatted correctly; it is distracting to the reader otherwise.
- Page 1 Lines 18-19: Please provide citations for this
- For the derivation of the uncertainty model, a table of all the variables and their definitions would be useful for the reader
- Page 9 Line 16: "Second partial derivative" is confusing here. Just reference equation 19 instead
- Figure 5: The lower right panel does not fit the lower left panel
- Figure 6: Equation number in figure does not match the caption
- Figure 7: Equation number in figure does not match the caption
- Table 2 & 3: It took me a while to figure out what I was looking at here. It could use some reformatting
- Figure 8: Check units
- Page 22 Lines 3-5: This isn't true in general, but is for the specific configuration tested
- Page 22 Lines 7-9: This isn't true in my experience

**References**

Stauffer, R., G. J. Mayr, M. Dabernig, and A. Zeileis, 2015: Somewhere Over the
     Rainbow: How to Make Effective Use of Colors in Meteorological Visualizations.
     *Bull. Amer. Meteor. Soc.*, **96**, 203–216,
     https://doi.org/10.1175/BAMS-D-13-00155.1.

---

## Author Comment (AC1) · 4 Mar 2021

Comments to Reviewer 1.

We thank the reviewer for the detailed comments. Of course we are saddened by the negative recommendation and would like to take this opportunity to refute the main criticisms. Should the editor in any case consider our paper for publication we will happily deal with the remaining issues kindly indicated by the reviewer.

The reviewer's main criticism is that we have adopted 'a very simplistic atmospheric model', in particular 1) that we have used a power-law shear model and 2) that we have not included horizontal gradient.

We strongly believe that the choice of power-law shear is in no way a weakness or limitation. What is required is a simple and approximate parametrization of how wind speed changes with height, including that the gradient depends on sensing height above the surface and having a parameter representing the strength of the shear. The power-law fulfills the role admirably with the advantage that the strength of the shear, the alpha parameter, is well known and typical values are readily available. This choice of model keeps the mathematics as simple as possible. A logarithmic model could also have been used, the mathematics would have been more complicated and we would have had to choose semi-arbitrary values for two or more parameters. Our estimate of the local wind speed gradient would not have been more accurate and therefore neither would our derived wind speed uncertainty estimates. A more sophisticated shear model is of course essential to describe wind speed variations over a significant height range but that is not our ambition here. We require simply a reasonable ('ball park') estimate of the vertical wind speed gradient so that we can in turn estimate the effect of elevation angle and range errors on the reconstructed wind speed. In a revision of this paper, we would of course include this justification and can readily see that it is missing in the current version.

The second main criticism concerns our assumption of horizontal homogeneity, i.e. not including any horizontal wind speed gradients. Again we regret the omission of a justification but we do not believe that this is a significant weakness of our model. The justification is simply that (in the typical wind energy applications that we have in mind at least) horizontal gradients are typically no greater than 1%/km whereas vertical shear is of the order 0.1%/m (e.g. measuring at 100m with an alpha=0.1), a factor 100 greater. Wind speed errors due to measuring at the wrong height (e.g. due to (inclined beam) range or elevation angle errors) are typically one or two orders of magnitude greater than measuring at the wrong horizontal position. Including typical horizontal gradients (e.g coastal gradients or gradients due to wind farm blockage) would only very exceptionally affect the uncertainty estimates. A clear exception would be measuring across wind turbine wakes. In such applications, a detailed uncertainty analysis is much less relevant than for example in determining a coastal wind resource estimate.

A third limitation seen by the reviewer is that *'the error in the observed radial speed is treated as a constant, when in fact it varies strongly with range (and depends on aerosol loading) in real lidar systems'*. We interpret this statement as saying that the statistical uncertainty (scatter) of the radial speed varies strongly with range and aerosol loading. This we partly agree with, although for signal levels well above threshold, the statistical uncertainty is small in comparison to the type B uncertainty (unknown bias) associated with the radial speed. This type B uncertainty is the uncertainty that we can

assign to the radial speed following a calibration. Typically this will essentially be the uncertainty of the reference cup anemometer in a field calibration. As stated previously, this will be much larger (typically 1-2%) than the random error and much more relevant for uncertainty estimates since no amount of averaging will reduce it (we are normally concerned with averages of between 30s and 10 mins). We do assume that this type B uncertainty remains constant with range since it is more a property of the calibration process than the lidar itself. Clearly, the ability of the lidar to return a valid signal decreases with range depending on the aerosol load. We do not treat this aspect in our model but it should clearly be included as an additional parameter in a campaign design. It would in any case be quite straightforward to use values of the los uncertainty that vary with range should this be relevant.

The reviewer does recognize the possible value of the model as a campaign planning tool but we take polite exception to the expression '*the only practical value*' since this is indeed the intended value and an extremely valuable one! For example, the model is currently being used to plan a major experiment to investigate global blockage at offshore wind farms. Here the uncertainty is fundamental to distinguishing between competing hypotheses and therefore determining the success of the project. We know of no other model that combines the necessary aspects of the uncertainty determination.

Our proposal for a significant improvement to the paper would be to add a section justifying and explaining our model assumptions and choices very much as we have outlined above. We hope that with this major improvement the reviewer will reconsider the recommendation against publication. If this approach is acceptable, we will of course consider and treat the reviewers other comments for which we are very grateful.

---

## Author Comment (AC2) · 4 Mar 2021

Comments to Reviewer 2.

We thank the reviewer for the detailed comments. Of course we are saddened by the negative recommendation and would like to take this opportunity to refute the main criticisms. Should the editor in any case consider our paper for publication we will happily deal with the remaining issues kindly indicated by the reviewer.

The reviewer believes that our modelling assumptions are too restrictive, stating *'There are numerous assumptions that must be made to solve the equations analytically.'* This statement is not strictly correct. We are not **solving** equations analytically, we are simply applying the GUM methodology which basically requires us to calculate and combine a number of (rather complicated) partial derivatives. We could have included a logarithmic vertical shear model rather than the power-law, we could have included horizontal shear parameters and we could have included correlations between the uncertainties. We choose not to do so because we believe that none of these would significantly change the result, only make the system of resulting equations even more complicated. There is nothing preventing us (or others, the code is open source and freely available) from extending the model as and where necessary, for example by adding horizontal shear gradients and including correlations between uncertainties if we identify this requirement.

The reviewer also states that our assumptions are '*incredibly limiting*'.  We do not believe this to be the case. There will be many relevant applications, for example flat terrain and offshore wind resource assessment, where the model is readily applicable. Here we will attempt to justify our reasoning:

1) Horizontal homogeneity – We have not included horizontal gradients of wind speed (or direction) since for the applications familiar to us (in wind energy) where accuracy of wind speed is an issue, the horizontal gradients may be of the order of 1%/km whereas the vertical gradients are of the order 0.1%/m, a factor 100 greater. It is the vertical shear that drives the wind speed error (through elevation angle and inclined beam range errors). Including realistic horizontal gradients would not significantly change our horizontal speed uncertainty estimates. This does not mean that we can not use the model for applications where horizontal gradients are present (measuring such gradients is a major and important use case). It simply reflects that the horizontal gradients themselves make no significant contribution to the uncertainty estimates since they are far too small. An example to illustrate this: A range error of 10m on a beam inclined at 1°, measuring at 100m in a vertical shear with power-law exponent 0.1 gives a speed error of about 0.17%  (sin1x10/100x0.1*100). The error due to the horizontal shear (1%/km) would be  10/1000*1% = 0.01%.

2) Power-law vertical wind speed profile – This is a deliberate choice. What is required is a simple and approximate parametrization of how wind speed changes with height, including that the gradient depends on sensing height above the surface and having a parameter representing the strength of the shear. The power-law fulfills the role admirably with the advantage that the strength of the shear, the alpha parameter, is well known and typical values are readily available. This choice of model keeps the mathematics as simple as possible. A logarithmic

model could also have been used, the mathematics would have been more complicated and we would have had to choose semi-arbitrary values for two or more parameters. Our estimate of the local wind speed gradient would not have been more accurate and therefore neither would our derived wind speed uncertainty estimates. A more sophisticated shear model is of course essential to describe wind speed variations over a significant height range but that is not our ambition here. We require simply a reasonable ('ball park') estimate of the vertical wind speed gradient so that we can in turn estimate the effect of elevation angle and range errors on the reconstructed wind speed.

3) Uncertainty is uncorrelated – Again in the spirit of keeping the model from exploding mathematically, we have assumed no correlation between uncertainty components and therefore use the simpler form of the GUM equation. Where correlations are identified, they can be added as necessary. There could be some arguments that the los speed uncertainties are correlated to some degree. Please see our comments later in this note.

4) Lidars use shallow elevation angles - Most applications of dual-Doppler lidars require only moderate elevation angles since otherwise vertical wind speed components can begin to corrupt the reconstructions. Typical uses for dual-Doppler lidar where uncertainty is important include determining wind resources. Almost invariably this will be measuring at ranges of some kilometers at heights of around 100m. These will be quite shallow angles.

5) Vertical velocities are minimal – Linked very closely to the previous assumption. In typical applications we need to use low elevation angles. In this case, vertical velocities are actually unimportant since their contribution will be very small. At slightly larger elevation angles, the model will still be acceptable provided that vertical velocities are insignificant. In typical flat terrain and offshore wind resource estimation, this will also be a fair assumption. It might begin to fail in measurements over complex terrain.

In a revision of our paper, we should clearly include these justifications as well as more critically examining the limits of application of our model.

The reviewer suggests a Monte Carlo simulation as a more appropriate approach. Whilst we have used a Monte Carlo simulation to check the veracity of this model, we do believe that our approach is highly relevant for campaign planning, where the analytical expressions allow a complete uncertainty map to be generated extremely quickly. A Monte Carlo simulation would require tens or hundreds of thousands of iterations for each and every grid point. This would be extremely expensive numerically and surely rather slow. Indeed, the model is currently being used to plan a major experiment to investigate global blockage at offshore wind farms. Here the uncertainty is fundamental to distinguishing between competing hypotheses and therefore determining the success of the project. We know of no other model that combines the necessary aspects of the uncertainty determination.

The reviewer also takes exception to our assumption that the LOS velocity is a constant, stating that this is actually *'a function of range and aerosol backscatter',* also stating that *'this also violates the assumption that the uncertainty contributors are uncorrelated.'* If the reviewer is thinking of the random error on the radial wind speed then we agree that this would typically be a function of range and aerosol

load but disagree that this would imply correlation between uncertainty contributors. Uncertainties can be equal but completely uncorrelated. Correlation requires a mechanism whereby an unknown error on one source necessarily implies a correlated unknown error on the other source.

However, we consider the main contribution to the los uncertainty to be the type B (unknown bias) resulting from the calibration process. This will typically be a standard uncertainty of between 1.5-2% , coming essentially from the reference uncertainty of the cup anemometer used in the calibration process. Since this is a property of the calibration process, it is reasonable to assume that it is constant with lidar range. Obviously approaching the limits of the lidar range, the lidar is unable to return a reliable speed and the uncertainty will become very high. This effect is not included in our uncertainty model but would naturally be an important parameter in any campaign design.

In our interpretation of the los speed uncertainty, the reviewer's claim that the los uncertainties are correlated may actually be true. If it is the same lidar type or calibrated (and explicitly corrected) at the same facility, then there is a possibility that both lines of sight may have the same unknown error to some degree.  This limitation should be explained in the paper and possibly modified in future versions of the model.

Our proposal for a significant improvement to the paper would be to add a section justifying and explaining our model assumptions and choices very much as we have outlined above. We hope that with this major improvement the reviewer will reconsider the recommendation against publication. If this approach is acceptable, we will of course consider and treat the reviewers other comments for which we are very grateful.